# Weighted Metamorphosis for registration of images with different topologies

Anton François[1,2], Matthis Maillard[2], Catherine Oppenheim[3], Johan Pallud[3], Isabelle Bloch[4,2], Pietro Gori[2], and Joan Glaunès[1]

[1] Université de Paris-Cité, Paris, France
[2] LTCI, Télécom Paris, Institut Polytechnique de Paris, France
[3] UMR 1266 INSERM, IMA-BRAIN, IPNP, Paris, France
[4] Sorbonne Université, CNRS, LIP6, Paris, France

**Abstract.** We present an extension of the Metamorphosis algorithm to align images with different topologies and/or appearances. We propose to restrict/limit the metamorphic intensity additions using a time-varying spatial weight function. It can be used to model prior knowledge about the topological/appearance changes (e.g., tumour/oedema). We show that our method improves the disentanglement between anatomical (i.e., shape) and topological (i.e., appearance) changes, thus improving the registration interpretability and its clinical usefulness. As clinical application, we validated our method using MR brain tumour images from the BraTS 2021 dataset. We showed that our method can better align healthy brain templates to images with brain tumours than existing state-of-the-art methods. Our PyTorch code is freely available here: `https://github.com/antonfrancois/Demeter_metamorphosis`.

**Keywords:** Image registration · Metamorphosis · topology variation · brain tumour

## 1 Introduction

When comparing medical images, for diagnosis or research purposes, physicians need accurate anatomical registrations. In practice, this is achieved by mapping images voxel wise with a plausible anatomical transformation. Possible applications are: computer assisted diagnosis or therapy, multi-modal fusion or surgical planning. These mappings are usually modelled as diffeomorphisms, as they allow for the creation of a realistic one to one deformation without modifying the topology of the source image. There exists a vast literature dealing with this subject. Some authors proposed to use stationary vectors fields, using the Lie algebra vector field exponential [1,2,14], or, more recently, Deep-Learning based methods [5,16,19,21,22,29]. Other authors used the Large Diffeomorphic Deformation Metric Mapping (LDDMM) that uses time varying vector fields to define a right-invariant Riemannian metric on the group of diffeomorphisms.

One advantage of this metric is that it can be used to build a shape space, providing useful notions of geodesics, shortest paths and distances between images [3,6,30,31]. A shortest path represents the registration between two images.

However, clinical or morphometric studies often include an alignment step between a healthy template (or atlas) and images with lesions, alterations or pathologies, like white matter multiple sclerosis or tumour. In such applications, source and target images show a different topology, thus preventing the use of diffeomorphisms, which are by definitions one-to-one mappings. Several solutions have been proposed in order to take into account such topological variations. One of the first methods was the Cost-Function Masking [7], where authors simply excluded the lesions from the image similarity cost. It is versatile and easy to implement, but it does not give good results when working with big lesions. Sdika et al. [24] proposed an inpainting method which only works on small lesions. Niethammer et al. proposed Geometric Metamorphosis [20], that combines two deformations to align pathological images which need to have the same topology. Another strategy, when working with brain images with tumours, is to use biophysical models [10,23] to mimic the growth of a tumour into an healthy image and then perform the registration (see for instance GLISTR [11]). However, this solution is slow, computationally heavy, specific to a particular kind of tumour and needs many different imaging modalities. Other works proposed to solve this problem using Deep-Learning techniques [8,12,15,25]. However, these methods strongly depend on the data-set and on the modality they have been trained on, and might not correctly disentangle shape and appearance changes.

The Metamorphic framework [13,27,30] can be seen as a relaxed version of LDDMM in which residual time-varying intensity variations are added to the diffeomorphic flow, therefore allowing for topological changes. Nevertheless, even if metamorphosis leads to very good registrations, the disentanglement between geometric and intensity changes is not unique and it highly depends on user-defined hyper-parameters. This makes interpretation of the results hard, thus hampering its clinical usage. For instance, in order to align a healthy template to an image with a tumour, one would expect that the method adds intensities only to create new structures (i.e., tumours) or to compensate for intensity changes due to the pathology (i.e. oedema). All other structures should be correctly aligned solely by the deformations. However, depending on the hyper-parameters, the algorithm might decide to account for morphological differences (i.e. mass effect of tumours) by changing the appearance rather than applying deformations. This limitation mainly comes from the fact that the additive intensity changes can theoretically be applied all over the image domain. However, in many clinical applications, one usually has prior knowledge about the position of the topological variations between an healthy image and a pathological one (e.g., tumour and oedema position).

To this end, we propose an extension of the Metamorphosis (M) model [13,27], called Weighted Metamorphosis (WM), where we introduce a time-varying spatial weight function that restricts, or limits, the intensity addition

only to some specified areas. Our main contributions are: 1./ A novel time-varying spatial weight function that restricts, or limits, the metamorphic intensity additions [13,27] only to some specified areas. 2./ A new cost function that results in a set of geodesic equations similar to the ones in [13,27]. Metamorphosis can thus be seen as a specific case of our method. 3./ Evaluation on a synthetic shape dataset and on the BraTS 2021 dataset [17], proposing a simple and effective weight function (i.e., segmentation mask) when working with tumour images. 4./ An efficient PyTorch implementation of our method, available at `https://github.com/antonfrancois/Demeter_metamorphosis`.

## 2   Methods

**Weighted Metamorphosis.** Our model can be seen as an extension of the model introduced by Trouvé and Younès [27,30]. We will use the same notations as in [9]. Let $S, T : \Omega \to [0, 1]$ be grey-scale images, where $\Omega$ is the image domain. To register $S$ on $T$, we define, similarly to [27,30], the evolution of an image $I_t$ ($t \in [0, 1]$) using the action of a vector field $v_t$, defined as $v \cdot I_t = -\langle \nabla I_t, v_t \rangle$, and additive intensity changes, given by the residuals $z_t$, as:

$$\dot{I}_t = -\langle \nabla I_t, v_t \rangle + \mu M_t z_t, \quad \text{s.t. } I_0 = S,\ I_1 = T,\ \mu \in \mathbb{R}^+. \tag{1}$$

where we introduce the weight function $M_t : x \in \Omega \to [0, 1]$ (at each time $t \in [0, 1]$) that multiplies the residuals $z_t$ at each time step $t$ and at every location $x$. We assume that $M_t$ is smooth with compact support and that it can be fully computed before the optimisation. Furthermore, we also define a new pseudo-norm $\| \bullet \|_{M_t}$ for $z$. Since we want to consider the magnitude of $z$ only at the voxels where the intensity is added, or in other terms, where $M_t(x)$ is not zero, we propose the following pseudo-norm:

$$\|z_t\|_{M_t}^2 = \left\| \sqrt{M_t} z_t \right\|_{L^2}^2 = \langle z_t, M_t z_t \rangle_{L^2} \tag{2}$$

This metric will sum up the square values of $z$ inside the support of $M_t$. As usual in LDDMM, we assume that each $v_t \in V$, where $V$ is a Hilbert space with a reproducing kernel $K_\sigma$, which is chosen here as a Gaussian kernel parametrized by $\sigma$ [18,28]. Similarly to [27,30], we use the sum of the norm of $z$ and the one of $v$ (i.e., the total kinetic energy), balanced by $\rho$, as cost function:

$$E_{\text{WM}}(v, I) = \int_0^1 \|v_t\|_V^2 + \rho \|z_t\|_{M_t}^2 dt, \quad \text{s.t. } I_0 = S,\ I_1 = T,\ \rho \in \mathbb{R}^+ \tag{3}$$

where $z$ depends on $I$ through Eq. 1. By minimising Eq.3, we obtain an exact matching.

**Theorem 1.** *The geodesics associated to Eq. 3 are:*

$$\begin{cases} v_t = -\frac{\rho}{\mu} K_\sigma \star (z_t \nabla I_t) \\ \dot{z}_t = -\ \nabla \cdot (z_t v_t) \\ \dot{I}_t = -\langle \nabla I_t, v_t \rangle + \mu M_t z_t \end{cases} \tag{4}$$

where $\nabla \cdot (v)$ is the divergence of the field $v$ and $\star$ represents the convolution.

*Proof.* This proof is similar to the one in [30], Chap. 12, but needs to be treated carefully due to the pseudo-norm $\|z_t\|_{M_t}^2 = \left\langle z_t, \frac{1}{\mu}(\dot{I}_t + v_t \cdot \nabla I_t) \right\rangle_{L^2}$. We aim at computing the variations of Eq. 3 with respect to $I$ and $v$ and compute the Euler-Lagrange equations. To this end, we define two Lagrangians: $L_I(t, I, \dot{I}) = E_{\text{WM}}(\bullet, v)$ and $L_v(t, v, \dot{v}) = E_{\text{WM}}(I, \bullet)$ and start by computing the variations $h$ with respect to $v$:

$$D_v L_v \cdot h = \int_0^1 \langle 2(K^{-1}v_t + \frac{\rho}{\mu}z_t \nabla I_t), h_t \rangle_{L^2} dt \tag{5}$$

Then, noting that $\nabla_v L_v = 2(K^{-1}v_t + \frac{\rho}{\mu}z_t \nabla I_t)$ and since $\nabla_{\dot{v}}L_v = 0$, the Euler-Lagrange equation is:

$$\nabla_v L_v - \dot{\nabla}_{\dot{v}} L_v = 0 \Leftrightarrow v_t = \frac{\rho}{\mu}K \star (z_t \nabla I_t) \tag{6}$$

as in the classical Metamorphosis framework [30]. Considering the variation of $I$, we have $D_I \left\| \sqrt{M_t} z \right\|_{L^2}^2 = \left\langle z_t, \frac{1}{\mu}v_t \cdot \nabla h_t \right\rangle_{L^2}$, thus obtaining:

$$D_I L_I \cdot h = 2 \int_0^1 \left\langle z_t, \frac{1}{\mu}\nabla h_t \cdot v_t \right\rangle_{L^2} dt = \int_0^1 \left\langle -\frac{2}{\mu}\nabla \cdot (z_t v_t), h_t \right\rangle_{L^2} dt \tag{7}$$

and $D_{\dot{I}} L_I \cdot h = \int_0^1 \langle \frac{2}{\mu}z_t, h_t \rangle_{L^2} dt$. We deduce that $\nabla_I L_I = \frac{2}{\mu}\nabla \cdot (z_t v_t)$ and as $\nabla_{\dot{v}} L_v = \frac{2}{\mu}z_t$, its Euler-Lagrange equation is:

$$\nabla_I L_I - \dot{\nabla}_{\dot{I}} L_I = 0 \Leftrightarrow \dot{z}_t = -\nabla \cdot (z_t v_t) \tag{8}$$

We can first notice that, by following the geodesic paths, the squared norms over time are conserved ($\forall t \in [0, 1], \|v_0\|_V^2 = \|v_t\|_V^2$) and thus one can actually optimise using only the initial norms. Furthermore, since $v_0$ can be computed from $z_0$ and $I_0$, the only parameters of the system are $z_0$ and $I_0$. As it is often the case in the image registration literature, we propose to convert Eq.3 into an unconstrained inexact matching problem, thus minimising:

$$J_{\text{WM}}(z_0) = \|I_1 - T\|_{L_2}^2 + \lambda \left[ \|v_0\|_V^2 + \rho\|z_0\|_M^2 \right], \quad \lambda \in \mathbb{R}^+, I_0 = S \tag{9}$$

where $I_1$ is integrated with Eq.4, $\|v_0\|_V^2 = \langle z_0 \nabla S, K_\sigma \star (z_0 \nabla S) \rangle$ and $\lambda$ is the trade-off between the data term (based here on a L2-norm, but any metric could be used as well) and the total regularisation.

**Weighted function construction.** The definition of the weight function $M_t$ is quite generic and could be used to register any kind of topological/appearance differences. Here, we restrict to brain tumour images and propose to use an evolving segmentation mask as weight function. We assume that we already

have the binary segmentation mask $B$ of the tumour (comprising both oedema and necrosis) in the pathological image and that healthy and pathological images are rigidly registered, so that $B$ can be rigidly moved onto the healthy image. Our goal is to obtain an evolving mask $M_t : [0,1] \times \Omega \to [0,1]$ that somehow mimics the tumour growth in the healthy image starting from a smoothed small ball in the centre of the tumour ($M_0$) and smoothly expanding it towards $B$. We generate $M_t$ by computing the LDDMM registration between $M_0$ and $B$. Please note that here one could use an actual biophysical model [10,23] instead of the proposed simplistic approximation based on LDDMM. However, it would require prior knowledge, correct initialisation and more than one imaging modality. The main idea is to smoothly and slowly regularise the transformation so that the algorithm first modifies the appearance only in a small portion of the image, trying to align the surrounding structure only with deformations. In this way, the algorithm tries to align all structures with shape changes adding/removing intensity only when necessary. This should prevent the algorithm from changing the appearance instead of applying deformations (i.e. better disentanglement) and avoid wrong overlapping between new structures (e.g. tumour) and healthy ones. Please refer to Fig.1 for a visual explanation.

## 3   Results and Perspectives

**Implementation details.** Our Python implementation is based on PyTorch for automatic differentiation and GPU support, and it uses the semi-Lagrangian formulation for geodesic shooting presented in [9]. For optimisation we use the PyTorch's Adadelta method.

**Synthetic data.** Here, we illustrate our method on a 300x300 grey-scale image registration toy-example (Fig. 1). We can observe the differences in the geodesic image evolution for LDDMM, Metamorphosis (M) and Weighted Metamorphosis (WM) with a constant and evolving mask. First, LDDMM cannot correctly align all grey ovals and Metamorphosis results in an image very similar to the target. However, most of the differences are accounted for with intensity changes rather than deformations. By contrast, when using the proposed evolving mask (fourth row), the algorithm initially adds a small quantity of intensity in the middle of the image and then produces a deformation that enlarges it and correctly pushes away the four grey ovals. In the third row, a constant mask ($M_t = M_1, \forall t \in [0,1]$) is applied. One can observe that, in this case, the bottom and left ovals overlap with the created central triangle and therefore pure deformations cannot correctly match both triangle and ovals. In all methods, the registration was done with the same field smoothness regularisation $\sigma$ and integration steps. Please note that the four grey ovals at the border are not correctly matched with LDDMM and, to a lesser extent, also with our method. This is due to the L2-norm data term since these shapes do not overlap between the initial source and target images and therefore the optimiser cannot match them.

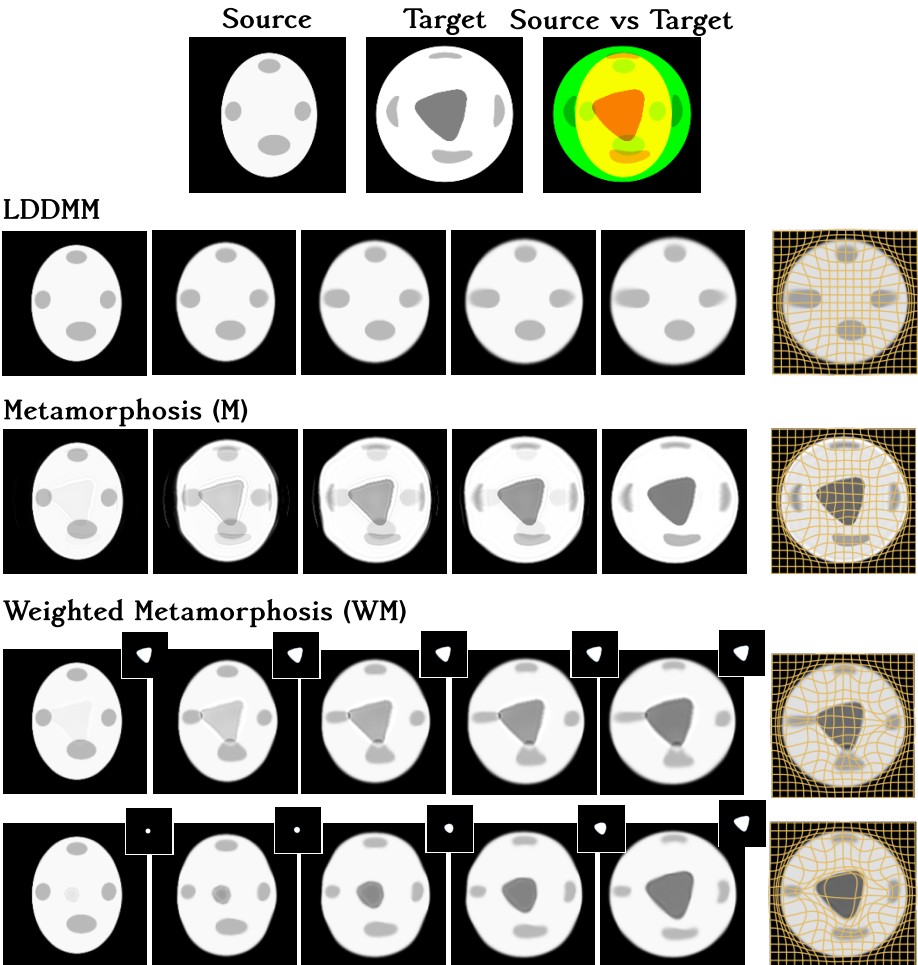

**Fig. 1. Comparison between LDDMM, Metamorphosis and our method.** Image registration toy example. Differently from the Source image (S), the Target image (T) has a big central triangle that has grown "pushing"' the surroundings ovals. Note that the bottom and left ovals in S overlap with the triangle in T. The two last rows show our method using a constant and time evolving mask (see Sec. 2). The used mask is displayed on the top right corner of each image. `see animations in GitHub in notebook : toyExample_weightedMetamorphosis.ipynb`

**Validation on 2D real data.** For evaluation, we used T1-w MR images from the BraTS 2021 dataset [4,17]. For each patient, a tumour segmentation is provided. We selected the same slice for 50 patients resizing them to 240x240 and making sure that a tumour was present. We then proceeded to register the healthy brain template SRI24 [26] to each of the selected slices (see Fig.2 for two examples). To evaluate the quality of the alignment we used three different

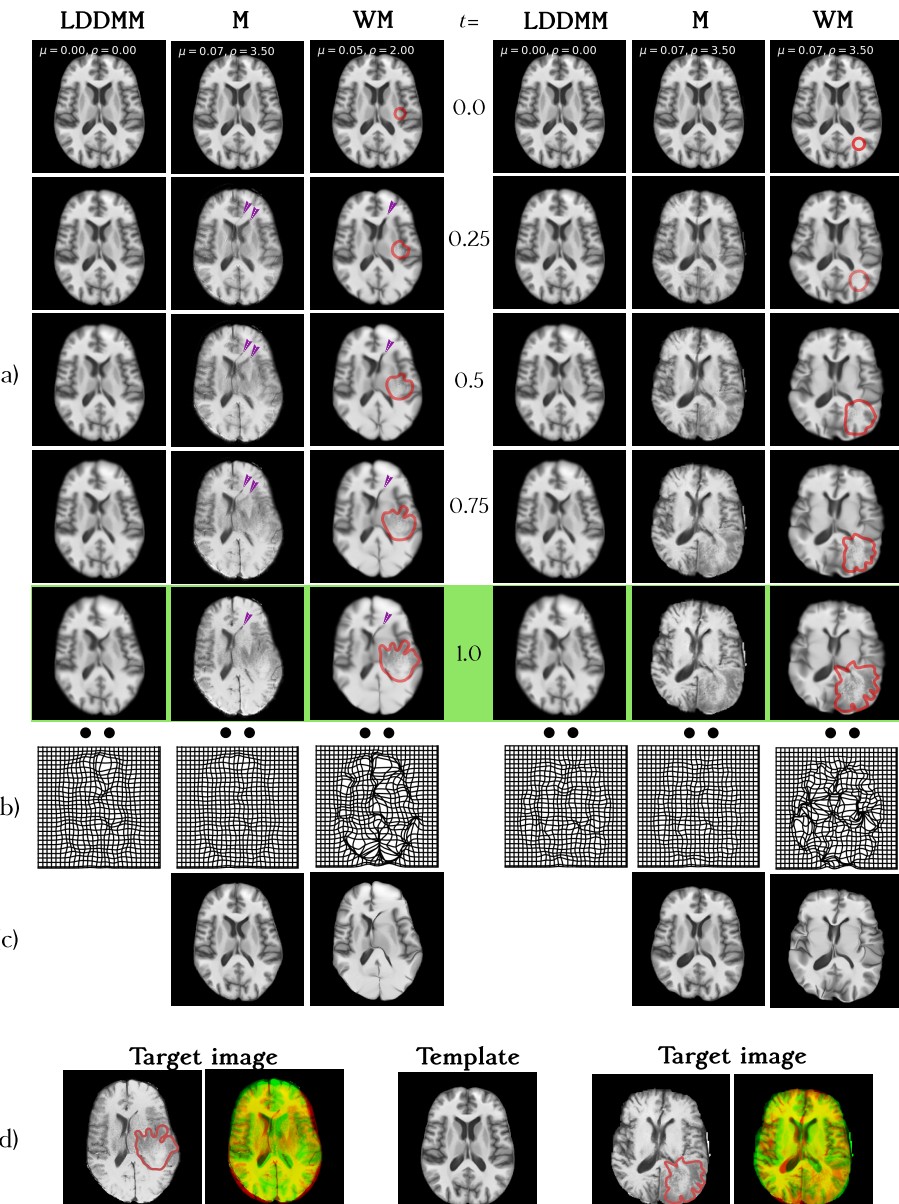

**Fig. 2. Registrations on MRI brains presenting brain tumours.** Two examples from BraTS database [4,17]. Comparison of geodesic shooting for LDDMM, Metamorphosis (M) and Weighted Metamorphosis (WM). (a&d) On the target images and the geodesic integration, the temporal mask is indicated by the red outline. The final result of each integration can be seen in the green outlined row. (b) The deformation grids retrieved from each method and (c) the template image deformed without intensity additions for each concerned method. Purple arrows in columns 2 and 3 in the top right part of each image show the evolution of one ventricle through registration: while M makes the ventricle disappear and reappear, WM coherently displaces the structure. (d) Target images with the segmentation outlined in red; the colored image is its superposition with the source. `see animations in GitHub in notebook : brains_weightedMetamorphosis.ipynb`

measures in Table 1: 1./ the Sum of Squared Differences (SSD) (i.e. L2-norm) between the target (T) and the transformed source (S) images. This is a natural choice as it is used in the cost function. 2./ the SSD between T and the deformed S without considering intensity changes. This is necessary since Metamorphoses could do a perfect matching without using deformations but only intensity changes. 3./ A Dice score between the segmentations of the ventricles in the deformed S and T. The ventricles were manually segmented. All methods should correctly align the ventricles using solely pure deformations since theses regions are (theoretically) not infiltrated by the tumour (*i.e.*, no intensity modifications) and they can only be displaced by the tumor mass effect.

**Table 1.** Quantitative evaluation for different registration methods. Results were computed on a test set of 50 2D 240x240 images from BraTS 2021 dataset. - ($*$) SSD for CFM is computed over the domain outside the mask.

| Method | LDDMM [18] | Meta. [9] | WM (ours) | MAE [8] | Voxelm. [5] | CFM [7] |
|---|---|---|---|---|---|---|
| SSD (final) | $223 \pm 51$ | $\mathbf{36 \pm 9}$ | $65 \pm 71$ | $497 \pm 108$ | $166.71 \pm 37$ | $49^* \pm 28$ |
| SSD (def.) | - | $112 \pm 21$ | $\mathbf{102 \pm 76}$ | $865 \pm 172$ | - | - |
| Dice score | $68.6 \pm 11.9$ | $74.1 \pm 9.3$ | $\mathbf{77.2 \pm 10.1}$ | $60.6 \pm 8.79$ | $66.8 \pm 10$ | $45.0 \pm 13.5$ |

We compared our method with LDDMM [6], Metamorphosis [27], using the implementation of [9], Metamorphic Auto-Encoder (MAE) [8], Voxelmorph [5] and Cost Function Masking (CFM) [7] (see Table 1). Please note that we did not include other deep-learning methods, such as [12,15], since they only work the other way around, namely they can only register images with brain tumours to healty templates. As expected, Metamorphosis got the best score for SSD (final) as it is the closest to an exact matching method. However, WM outperformed all methods in terms of Dice score obtaining a very low SSD (both final and deformation-only). This means that our method correctly aligned the ventricles, using only the deformation, and at the same time it added intensity only where needed to globally match the two images (i.e., good disentanglement between shape and appearance).

**Perspectives and conclusion.** In this work, we introduced a new image registration method, Weighted Metamorphosis, and showed that it successfully disentangles deformation from intensity addition in metamorphic registration, by using prior information. Furthermore, the use of a spatial mask makes our method less sensitive to hyper-parameter choice than Metamorphosis, since it spatially constrains the intensity changes. We also showed that WM improves the accuracy of registration of MR images with brain tumours from the BRATS 2021 dataset. We are confident that this method could be applied to any kind of medical images showing exogenous tissue growth with mass-effect. A future research direction will be the integration of methods from topological data analysis, such as persistent homology, to improve even more the disentanglement between geometric and appearance changes. We also plan to adapt our method to 3D data.

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
