# OpenReview forum: "Weighted Metamorphosis for registration of images with different topology."
_WBIR.info/2022/Workshop/Biomedical_Imaging_Registration — WBIR 2022_

### Official Review · Reviewer_bhSP · 2022-02-14

**Rating:** 3
**Confidence:** 5

**Deanonymize Review:**

no

**Detailed Comments:**

Equation 9 should mention that this is optimized subject to the Equation set (4).

Page 2: “... to mimic the growth of a tumour into an healthy image …” -> “... to mimic the growth of a tumour in a healthy image …”

Page 2: “ … a new cost function that brings to a set … ” -> “ … a new cost function that results in a set … ”

Page 8: “ … we measured the overlapping … “ -> “ … we measured the overlap …”

The bibliography shows many inconsistent capitalizations. For example “brats” instead of “BRATS” or “em” instead of “EM”.


**Paper Type:**

methodological development

**Strengths Weaknesses:**

This work proposes an extension of an approximate metamorphosis model. In particular, a time-evolving mask is added which only allows for changing mass at locations where the mask is non-zero. This time-dependent mask is pre-computed by LDDMM registration between an initial circle/sphere and the target mask. Qualitative synthetic results are shown as well as results based on 2D slices of the BRATS dataset. Overall, this is an interesting, simple idea. However, shortcomings are in the evaluation (making it unclear how well the method performs in practice) as well as the somewhat ad-hoc way of constructing the time-dependent mask.

Comments:

1. The evolution of the mask is computed by an independent LDDMM registration between a smoothed ball in the center of the tumor and a binary segmentation mask (presumably in the space of the target image). This seems to be a relatively arbitrary design choice. In particular, it does not seem to account for the movement of the underlying space as governed by Equation (4). Would it be possible to more directly couple these evolutions? And how does the current model behave in the context of pure infiltration or pure mass effect?

2. While the proposed model indeed offers more flexibility than the classical metamorphosis model, it seems like it still does not entirely get around the balancing issue between the metamorphosis term and the more standard LDDMM-like deformation. I.e., as \rho goes to zero I would expect very strong deformations and as \rho goes to infinity I would still expect most of the changes to be explained by the metamorphosis terms (at least in roughly the areas of the tumor, for example, as only there M>0).

3. It might be useful to show boxplots for the results. While the mean values for SSD and Dice for WM indeed seem to be the best, they show very high standard deviations in comparison to standard metamorphosis. Is there a good intuition for this effect?

4. It is unclear what the take-away from Figures 1 and 2 is. For the synthetic experiment (Fig. 1) it might have been useful to either define a synthetic ground truth deformation which could then be quantitatively assessed or some other quantitative evaluation measure. The qualitative evaluation is not very informative. For Fig. 2 the problem seems to be that while WM obtains better quantitative measures (see Table 1) the shown transformations (row b of Fig. 2) are widely different between the methods. This might be expected to a certain extent, but it appears troublesome that WM seems to show a highly irregular transformation across the entire slice. It might be useful to choose parameters that are method-specific, for example, based on cross-validation.

---

### Official Review · Reviewer_Yg7b · 2022-02-16

**Rating:** 5
**Confidence:** 4
**Recommendation:** Long Oral, Short Oral

**Deanonymize Review:**

no

**Detailed Comments:**

strengths:
- motivation is very nice and compact.
- the paper is very well written and the method clearly outlined
- evaluation is nicely done on a toy dataset and BraTS longitudinal data


weaknesses:
- "for diagnosis or research purposes, physicians need accurate anatomical registrations" -- at least for diagnosis this is not done. Images are compared side-by-side
- "in many clinical applications, one usually has prior knowledge
about the position of the topological variations between an healthy image and a
pathological one (e.g., the position of a tumour and oedema or of a lesion)." -- do the authors propose manual input or an automated segmentation model
- it seems a bit like a minor extension to [28, 13] and [7]. Essentially the weights are a semantic segmentation mask.
- p2: "that brings to a set of geodesic equations " -- what does this mean?
- why isn't the topology characterised explicitly, e.g. through persistent homology, since the weight function is anyway a segmentation mask? Just asking, I might have missed it in the paper!
- The toy example in Figure 1 is very nice, but doesn't the overlap of the triangle with the ovals violate assumptions about diffeomorphism? Results with the growing mask are nice but I guess this would be difficult to achieve in a real setup (as you wrote about the biophysical model, but this is a catch-22, since registration methods are also used to characterise e.g. the growth of tumours and I am not aware of good biophysical models that would provide accurate predictions of tumour growth perhaps even time reversed to get to a starting point.) As you show in Fig. 2 it can be done when longitudinal data is available.

**Paper Type:**

methodological development

**Strengths Weaknesses:**

The paper presents an extension of the Metamorphosis algorithm to align images with a different topology and/or appearance. The authors claim that their approach disentangles shape from topology. Evaluation is done on the BraTS 2021 dataset.


strengths:
- nicely written
- sound method
- well evaluated
- nice toy dataset
- good motivation

weaknesses:
- potentially minor extension of existing methods
- some questions regarding modelling temporally evolving segmentation masks

---

### Official Review · Reviewer_6TDL · 2022-02-19

**Rating:** 5
**Confidence:** 3
**Recommendation:** Long Oral

**Deanonymize Review:**

no

**Detailed Comments:**

- It would be helpful to improve the caption for Figure 2 - row d) was not explained, and why does c) contain only two images?
- In Figure 2, is it possible to see how the deformation fields evolve as well? (for at least the proposed method, perhaps another column could be added, or separate figure created)
- Please include a bit more background information on the dataset used
- Table 1 shows that your method produces a Dice score of 77.2%. Is this acceptable for the field? Are the results significant when compared to the other registration methods?
- Table 1 shows the effect of registration on the ventricles. What would happen if other brain structures were compared?


Overall, this was a well-written manuscript with potential for use in other types of data. It would be helpful to learn about potential limitations of the proposed method as well as clear insight into future work and challenges that the authors may encounter.


**Paper Type:**

methodological development

**Strengths Weaknesses:**

Strengths:
- The introduction is well-written and provides a good background of previous work and the motivation for the proposed approach, as well as the contributions of the authors
- The code is freely available
- The experiments performed for the results are ideal – as the authors perform experiments using both synthetic data and MR brain data.

Weaknesses:
- The limitations of the proposed method were not directly addressed and discussed
- The future work could be improved upon – what other types of data could this method be applied to? Are there plans to test on more data? - What would be the challenges in adopting the method to 3D and the integration of methods from topological data analysis?

---

### Decision · Program_Chairs · 2022-02-22

Accept